# Emerging Technologies for the Extraction of Marine Phenolics: Opportunities and Challenges

**DOI:** 10.3390/md18080389

**Published:** 2020-07-27

**Authors:** Adane Tilahun Getachew, Charlotte Jacobsen, Susan Løvstad Holdt

**Affiliations:** National Food Institute, Technical University of Denmark, Kemitorvet Building 204, 2800 Kgs Lyngby, Denmark; atige@food.dtu.dk (A.T.G.); suho@food.dtu.dk (S.L.H.)

**Keywords:** marine phenolics, emerging technologies, extraction

## Abstract

Natural phenolic compounds are important classes of plant, microorganism, and algal secondary metabolites. They have well-documented beneficial biological activities. The marine environment is less explored than other environments but have huge potential for the discovery of new unique compounds with potential applications in, e.g., food, cosmetics, and pharmaceutical industries. To survive in a very harsh and challenging environment, marine organisms like several seaweed (macroalgae) species produce and accumulate several secondary metabolites, including marine phenolics in the cells. Traditionally, these compounds were extracted from their sample matrix using organic solvents. This conventional extraction method had several drawbacks such as a long extraction time, low extraction yield, co-extraction of other compounds, and usage of a huge volume of one or more organic solvents, which consequently results in environmental pollution. To mitigate these drawbacks, newly emerging technologies, such as enzyme-assisted extraction (EAE), microwave-assisted extraction (MAE), ultrasound-assisted extraction (UAE), pressurized liquid extraction (PLE), and supercritical fluid extraction (SFE) have received huge interest from researchers around the world. Therefore, in this review, the most recent and emerging technologies are discussed for the extraction of marine phenolic compounds of interest for their antioxidant and other bioactivity in, e.g., cosmetic and food industry. Moreover, the opportunities and the bottleneck for upscaling of these technologies are also presented.

## 1. Introduction

Phenolic compounds are secondary metabolites produced by plants, microorganisms, and algae. The marine environment is an excellent source of bioactive compounds, and this includes phenolic compounds. There are many theories about why marine plant (seaweeds) tissues produce and accumulate phenolic compounds, and one of the most widely accepted theories states that phenolic compounds are produced as a defense mechanism to protect from biotic and abiotic factors. To adapt and survive in a very competitive and challenging marine environment, seaweeds (macroalgae) need to produce defense mechanisms from their metabolic pathways that protect them against the harsh environment and biological agents like UV protectant, anti-herbivory, and antioxidant [1]. The most pronounced environmental stress is that seaweeds often grow in intertidal locations where they are exposed to a frequently changing environment with high fluctuation of sunlight and oxygen, demanding from them a strong antioxidative defense system [1,2]. Nevertheless, the type and level of compounds vary from species to species, e.g., the geographic location, the growing season, and the environmental stress [3,4]. For instance, phlorotannins are phenolic compounds that are exclusively found in brown seaweeds [5], which grow in the mentioned harsh environments, and their concentration is highly variable across the different seasons [6,7].

Several types of phenolic compounds were isolated and characterized from brown, green, and red seaweed. Both the type and composition of the phenolic compounds vary across the different genera of seaweed [8]. The most commonly reported phenolic compounds are simple phenolic compounds like gallic acid, epicatechin, epigallocatechin, catechin gallate, protocatechuic acid, hydroxybenzoic acid, chlorogenic acid, caffeic acid, and vanillic acid [9,10]. Several flavonoids like quercetine, hesperidin, myricetin, and rutin were also reported in marine seaweeds [10,11,12,13] (Figure 1). Phlorotannins are the most commonly reported complex phenolic compounds in several brown seaweed species. Phlorotannins are oligomers of a phloroglucinol (1,3,5-trihydroxy benzene) monomer unit, with a large molecular size ranging from 126 Da to several thousand Da. Depending on the nature of the linkage between the aromatic units, there are four different classes of phlorotannins. When the linkage between the phloroglucinol unit is aryl-ether (C-O-C), they are called phloroethols and fuhalols; those with phenyl linkage (C-C) are fucols; fucophloroethols have both phenyl and aryl-ether linkages, and ecols with a benzodioxin linkage [14] (Figure 2). As secondary metabolites, phlorotannins play a series of recognized roles in the cell, such as anti-herbivory defense, antifouling activity, and UV protectants [15].

Marine phenolics show several biological activities [16], such as antioxidants [10,17], anti-inflammation [18,19,20], antimicrobial [21,22,23], anticoagulant [24], plant growth promoters [25], anticancer agents [26,27], angiotensin I-converting enzyme (ACE) inhibiting activity [28], antidiabetic [29], and antiproliferative [30]. To get all these benefits of the marine phenolics and use them in different delivery systems, such as food, pharmaceuticals, and cosmetics, the compounds need to be extracted from the sample matrix, characterized, and purified. Traditionally, marine phenolic compounds were extracted from marine resources using organic solvents. However, this extraction method had several drawbacks, including the use of a large volume of solvents, long extraction time, low extraction yield, degradation of extracted compounds, and difficulty of separating the extract from the solvent. In order to mitigate these challenges, several newly emerging technologies were used to extract phenolics from marine resources. Therefore, this review aimed to summarize the most frequently used emerging and sustainable technologies in the extraction of marine phenolics from seaweed. Moreover, the opportunities and challenges are discussed for the upscaling of these technologies.

## 2. Traditional Extraction of Marine Phenolics

Traditional extraction, commonly known as solvent extraction or solid–liquid extraction (SLE), is the most commonly and frequently used extraction technique. SLE can be done in several ways, such as; refluxing using soxhlet; boiling the sample and solvent with or without stirring for a certain duration; or maceration with continuous stirring (often for long duration, and sometimes even overnight). Several kinds of solvents and a mixture of solvents with wide polarity ranges are used to extract marine phenolics. The solvents include methanol, ethanol, acetone, ethyl acetate, trichloromethane, a mixture of water and organic solvents like ethanol, acetone, and acetonitrile, and methanol, at different mixing ratios [31,32,33,34,35,36]. When a mixture of solvents like water and ethanol is used, the extraction yield of phenolic compounds is reported to be higher than each of the solvents used individually, due to different optimums of extractability of compounds of different polarities, for the different solvents in a mix [37]. However, conventional extraction methods usually involve the use of a large volume of solvents, longer extraction time, and high temperature. Such harsh extraction conditions lead to the possibility of oxidation and hydrolysis of the phenolic compounds. Moreover, the upscaling of this technology at an industrial level would be difficult, owing to practicality, energy, economic, and environmental considerations [38]. Thus, to overcome the challenges associated with the method, several newly emerging extraction technologies were introduced.

## 3. Emerging Technologies for Extraction of Marine Phenolics

The newly emerging technologies could be based on the energy/mechanism they use for the extraction, such as ultrasound-assisted extraction (UAE); microwave-assisted extraction (MAE); based on the use of biological agents such as enzymes, enzyme-assisted extraction (EAE); based on the use of new type of solvents like pressurized solvent extraction (PLE); and supercritical fluid extraction (SFE). There are also less frequently used technologies, such as pulsed electric field-assisted extraction (PEF) and ohmic heating, where an electrical current is passed through the material and generates heat. Moreover, in recent years, there are new types of solvents, which are increasingly used for the extraction of phenolic compounds, and these solvents include, ionic liquids and deep eutectic solvents utilizing the different melting points of the constituents. Additionally, two or more of these methods were also used in combination [39,40]. Recent advances in the use of these extraction methods to extract marine phenolics from seaweeds are discussed below.

### 3.1. Enzyme-Assisted Extraction (EAE)

EAE offers several advantages, as compared to conventional extraction methods, including low operation temperature and the use of environment-friendly solvent. EAE is not a new extraction method to recover bioactives from a marine organism, however, ever-increasing research in the extraction process optimization and intensification coupled with a constant study in the discovery of new robust enzymes makes it an emerging technology.

Phenolic compounds can exist in several forms in the plant, microorganism, and algal material, such as in free soluble form or as insoluble complex forms [41]. Most marine phenolic compounds either make a complex with other macromolecules like protein and carbohydrate, or inside the cell, surrounded by a thick cell wall polysaccharide like alginate. Some phenolics are, however, easily accessible when located in the outer cell walls in physodes [8,42]. The discovery and application of cell wall degrading enzymes are based on their ability to degrade polysaccharides like alginate. For instance, phlorotannins, the most abundant form of phenolic compounds in brown seaweeds, usually form a covalent bond with proteins to produce a protein–polyphenol complex. To release these bound phenolic compounds, and make them available for extraction, an enzyme that is capable of hydrolyzing the protein–polyphenol bond is required [41]. In recent years, several novel enzymes were discovered from marine organisms. Most of the enzymes are cell wall polysaccharides degrading enzymes like alginate-lyases, glucuronan-lyases, carrageenases, laminarinases, and agarases [43]. When such structural and cell wall polysaccharides are degraded, the contents of the cell, including the phenolic compounds are released into the extraction medium, and thus increases the extraction yield. Some recently discovered enzymes had a higher activity, as compared to the commercially available enzymes. Ihua et al. [44] used extracts from enzyme-active bacterial isolate, which was obtained from decaying *Ascophyllum nodosum* for EAE of phenolic compounds from *Fucus vesiculosus*. They reported that the yield of phenolic compounds extracted, even at a low temperature of 28 °C, was higher than all three types of commercial enzymes (cellulase, proteases, and xylanase) used in the study. From the three commercial enzymes studied, xylanase yielded the best phenolic compounds (35.7 mg PGE/g dry weight (DW)), whereas enzyme active bacterial isolate yielded 44.8 mg PGE/g DW. However, most of such enzymes are isolated and tested at laboratory scale and are not currently available for industrial applications [45]. Thus, this research indicated that new types of robust and efficient enzymes are yet to be discovered from the marine environment and more integrated research is needed to accomplish this.

Although there is a problem in attacking specific cell wall polysaccharides, nonspecific enzymes that can degrade the polysaccharide and protein are available in large quantities for use in labs, pilots, and even at an industrial scale. Several researchers reported the use of such enzymes to recover marine phenolics from seaweed (Table 1). As shown in Table 1, the yield of phenolic compounds was affected by different factors, including the type of enzyme used, the enzyme concentration, temperature, the pH of the media, and the treatment duration. The type of enzymes used for the extraction process is very important. Generally, when comparing the yield of phenolic compounds produced by carbohydrases and proteases, most reports showed that those treated with carbohydrases had higher phenolic compounds [39,46,47]. The reason could be the differences in their attacking targets. Carbohydrases attack the cell wall polysaccharides and release the cell contents into the extraction medium to increase the extraction yield of the phenolic compounds. In contrast, proteases attack the protein and increase the concentration of protein in the extraction medium, which might subsequently react with phenolic compounds to form a protein–polyphenol complex in the extraction medium. The produced complex eventually makes aggregates and precipitates, and thereby results in a possible reduction of the phenolic compounds extraction yield [48]. Therefore, a careful selection of enzymes and optimizing the extraction parameters is a fundamental part of EAE. EAE is reported by itself or with the combination of other types of extraction technologies like UAE, MAE, PLE, and SFE. When EAE is combined with PLE and SFE, the pretreatment of biomass, together with the enzyme degrades the cell wall, and subsequently, the phenolic compounds are easily available for extraction by the solvents [48].

### 3.2. Ultrasound-Assisted Extraction (UAE)

Ultrasound-assisted extraction is an emerging potential technology that can accelerate heat and mass transfer and was successively used in extraction. Ultrasound waves alter the physical and chemical properties after interaction with the exposed material. The cavitational effect of the ultrasound waves facilitates the release of extractable compounds, and furthermore, enhances the mass transfer by disrupting the plant cell walls. UAE is a clean method that avoids the use of a large quantity of solvent, along with cutting down in the working time. Ultrasounds are successfully employed in the extraction field [49,50], and is well-known to have a significant effect on the rate of various processes in the chemical and food industry. Much attention is given to the application of ultrasound for the extraction of natural products that typically need hours or days to reach completion with conventional methods. With the use of ultrasound, full extractions can now be completed in minutes, with high reproducibility, reducing the consumption of solvent, simplifying manipulation and work-up, and yielding higher purity of the final product. This also eliminates post-treatment of wastewater and consumes only a fraction of the fossil energy normally needed for a conventional extraction method like the Soxhlet extraction, maceration, or steam distillation. Several classes of food components like aromas, pigments, polyphenols, and other organic and mineral compounds were extracted and analyzed efficiently from a variety of matrices, such as oils from garlic, phenolics from citrus peel and wheat bran, aromas from tea, and lycopene/pigment from tomatoes [49].

In more detail, the UAE is based on cavitation, i.e., the creation, growth, and sudden implosion (collapse) of bubbles with tremendous energy release from each of them, sometimes referred to as hot spots [51,52]. In the bulk solvent, the bubble collapse is symmetrical, but near to the surface, it is asymmetrical and generates a high-speed jet of liquids. In the case of cavitation in an extraction system, the jet hits the surface of the sample matrix and provides a continuous circulation of new solvents at the surface, producing deep penetration of solvent into the sample particle, continuous solvent mixing, and sometimes particle size reduction. The deep penetration of the solvent coupled with size reduction, enhances the extraction efficiency of the phenolic compounds [53]. UAE is used to extract phenolic compounds from marine resources. Several researchers extracted phenolics from brown algae (*Phaeophyceae*); *Ascophyllum nodosum* [54], *Laminaria japonica*, *Fucus vesiculosus* [55], green algae (*Chlorophyceae*); *Codium tomentosum* [39], and the red alga (*Rhodophyceae*); *Laurencia obtusa* [56] (Table 2). The extraction yield of phenolic compounds using UAE could be significantly affected by several experimental parameters. The parameters include the ultrasound energy/power, the sample-to-solvent ratio, a combined ratio of different solvents, the extraction temperature, time, and the particle size of the sample. To investigate the effect of each parameter, and the combination of two or more parameters on the extraction efficiency of marine phenolics using UAE, several statistical models were evaluated.

In their optimization study for extraction of phenolic compounds from red seaweed *L. obtusa* using UAE, Topuz et al. [56] determined the optimum extraction conditions of solvent—seaweed ratio, 24.3:1; extraction temperature, 45.3 °C; and the extraction time of 58 min, to recover a maximum yield of TPC 26.2 mg GAE/g seaweed. Dang et al. [57] also optimized the UAE of phenolic compounds from brown seaweed *Hormosira banksii*, using response surface methodology, and reported that the optimum conditions for maximum phenolic content were temperature, 30 °C; time, 60 min; and power, 150 W. According to this study, the application of UAE at the optimal conditions increased the yield of TPC by 143%, as compared to the conventional extraction method. In another study, Vázquez-Rodríguez et al. [58] optimized the extraction parameters, namely, extraction temperature (50–65 °C), ultrasound power density (1.2–3.8 W cL^−1^), solvent/seaweed ratio (10–30 mL g^−1^), and ethanol concentration (25–100% ethanol in water), to recover phlorotannins from brown seaweed *Silvetia compressa*. They reported a maximum of 7.3 mg PGE/g dry seaweed at (X1 = 50 °C, X2 = 3.8 W cL^−1^, X3 = 30 mL g^−1^ seaweed meal, and X4 = 32.3%). Moreover, the study showed that the ultrasound power density was the most influential parameter for the extraction of the phlorotannins.

### 3.3. Microwave-Assisted Extraction (MAE)

Microwave heating or microwave-assisted extraction (MAE) is used extensively to extract valuable materials from plant and animal resources. In details, microwave heating is generated by dipole rotation of a polar solvent and ionic condition of dissolved ions and this rapid volumetric heating leads to effective cell rupture, releasing the compounds into the solvent. MAE is used to extract phenolic compounds from several seaweed species (Table 3). Yuan et al. [59] applied microwave irradiation at 110 °C for 15 min, to extract phenolic compounds from four economically important brown seaweed species *Ascophyllum nodosum*, *Laminaria japonica*, *Lessonia trabeculate*, and *Lessonia nigrecens*. They also conducted the conventional method, agitation at room temperature for 4 h, for comparison. They found a TPC concentration of 139.8, 73.1, 74.1, and 107.1, GAE mg/100 g dry weight of each seaweed, respectively, which were much higher than the TPC obtained through the conventional extraction technique of 51.5, 38.5, 49.8, and 78.1 GAE mg/100 g DW, respectively, for the same set of seaweed samples.

The advantage of MAE over the conventional method, is getting a higher amount of TPC at a very short time, at a temperature, as high as 110 °C, which normally degrades and reduces the yield of phenolic compounds. Manusson et al. [60] conducted a comparison study of MAE extraction with conventional solid–liquid extraction of phloroglucinol from several brown seaweed species, and reported that the yield of polyphenols using MAE increased up to 70%, as compared to SLE. According to this study, the high amount of phenolic compounds in MAE, as compared to SLE is due to the accessibility of up to 40% of cell-wall bound polyphenols by MAE, which was not possible with SLE. Li et al. [61] optimized MAE of phenolic compounds from green seaweed species *Caulerpa racemose*, using an L_18_(3)^5^ orthogonal experimental array. In this study, a maximum of 67.9 mg GAE/100 g dried sample was obtained at optimum conditions of microwave power, 200 W; ethanol concentration, 60%; extraction time, 40 min; extraction temperature, 50 °C; and solvent-to-material ratio, 40 mL/g. In general, the efficiency of MAE was affected by several factors, including the microwave power, the type of solvents, the composition of solvents used, extraction time, temperature, and the sample-to-solvent ratio. Microwave power was found to be effective in increasing the extraction yield. However, too much power, on the contrary, would result in overheating of the system and lead to the degradation of heat-labile compounds such as polyphenols. The effect of temperature was also observed to be similar to microwave power. Another important factor affecting the yield of MAE is the type of solvent used in the extraction process. Water is a good solvent to absorb the microwave energy, and distribute the energy evenly in the extraction medium. However, the solubility of phenolic compounds in water is low, as compared to other organic solvents like ethanol. To overcome such problems, several researchers tested a mixture of water and ethanol in different mixing ratios. Several studies optimized the extraction parameters using statistical models, so that they could get the maximum polyphenol compounds with a possible combination of extraction parameters [26,27,61,62,63,64].

### 3.4. Pressurized Liquid Extraction (PLE)

Pressurized liquid extraction (PLE) is also known as accelerated solvent extraction (ASE) or subcritical water extraction (SWE), when the solvent used is only water in a temperature range of boiling point of water (100 °C) and critical temperature of water (374 °C). This is a newly emerging technology for the extraction of marine phenolics. This technology has several advantages over many other extraction technologies. This extraction takes a very short time, uses a relatively lower amount of solvents, and hence requires minimum consumption of solvent, and since the extraction is conducted in the absence of light and oxygen, the degradation of phenolic compounds is very low [67]. For SWE, we can easily modify the polarity of water by tuning the extraction temperature. For instance, increasing the temperature of water from room temperature to 200–250 °C, reduces the dielectric constant of water, which is a measure of the polarity of water from 80 to 30–25. This value is close to the dielectric constant of those of organic solvents like ethanol and methanol. These properties of water, at subcritical conditions, would enable water to extract some of the organic compounds that are otherwise not extractable by water at normal conditions. Thus, this could make SWE, the greenest process of all PLEs [68].

In recent years, several researchers reported the extraction of phenolic compounds from a variety of seaweed species, using SWE (Table 4). Different solvents such as ethanol, hexane, ethyl acetate, acetone, and a mixture of different solvents, like water and ethanol at different ratios, were used to extract marine phenolics from seaweed [69]. The extraction yield of phenolic compounds using PLE is affected by numerous factors, including temperature, pressure, the type of the solvent, the ratio between the sample and solvent, the extraction time, and the particle size of the sample. Among these factors, the temperature is by far the most influential one determining the extraction yield. Increasing the temperature reduces the viscosity and surface tension, and increases the diffusivity of the solvent, which consequently increases the mass transfer of the solvent into the sample matrix, to enhance the extraction yield. However, high extraction temperature might not always favor the extraction yield. High temperatures could potentially lead to decomposition of heat-sensitive compounds like phenolics and hence reduce the yield [70]. Pangestuti et al. [71] investigated the effect of temperature (120–270 °C), solid-to-liquid ratio (1:150 to 1:50), and static extraction time of 10 min, on total phenolic content of tropical red seaweed *Hypnea musciformis* extracts, obtained using SWE. In this study, the authors observed that increasing the solid-to-liquid ratio, increased the TPC at all extraction temperatures. Similarly, increasing temperature, also linearly increased the TPC content until it reached 210 °C, and after 210 °C, the TPC started to decline. The possible reason mentioned in this study for the reduction of TPC at higher temperatures was the thermal degradation of the phenolic compound beyond 210 °C. The maximum TPC reported in this study was 39.75 ± 0.2 mg GAE/g dried seaweed and this was recorded at solid-to-liquid ratio of 1:50 and a temperature of 210 °C. In another study, Gereniu et al. [72] studied pressurized hot water extraction (PHWE) of *Kappaphycus alvarezii* at different temperatures (150 to 300 °C) and pressure (1 MPa to 10 MPa), and reported the maximum TPC content at 270 °C and 8 MPa.

The type of solvent or mixture of solvents and their polarity is another type of very important parameter that significantly affect the yield of phenolic compounds. Otero, López-Martínez, and García-Risco [69] applied four different solvents that had wide polarity ranges, like hexane, ethyl acetate, ethanol, and ethanol:water (1:1), to extract phenolic compounds from brown alga *Laminaria ochroleuca*, using PLE. They reported that PLE extract with ethanol:water (1:1, *v*/*v*) showed the highest 173.7 mg GAE/g extract TPC, whereas the hexane extract showed the least TPC, with only 6 mg GAE/g extract. This indicated that careful selection of solvent is very important in PLE to get the best possible extraction yield and quality of phenolic compounds.

### 3.5. Supercritical Fluid Extraction (SFE)

A supercritical fluid is any substance that is kept at a temperature above its critical point (Tc) and critical pressure (Pc). When a substance is kept at such conditions, it has different physical and thermodynamic properties. The physical properties like viscosity, diffusibility, dielectric constant, density, and surface tension, all significantly change, compared to the substance at its standard atmospheric conditions. Moreover, we can tune these physical properties by changing the temperature and pressure. The most commonly used supercritical fluid in the food, pharmaceutical, and cosmetic industries is carbon dioxide (ScCO_2_). This is because carbon dioxide has a low critical temperature (31.4 °C) and pressure (73.8 bar), is not toxic, is easily available at high purity, and it is very easy to separate the gas from the extract [74]. ScCO_2_ can be used as the only solvent to extract marine phenolics or in a combination of organic solvents as entrainer. Several researchers reported the extraction of phenolic compounds from marine resources using ScCO_2_ [75,76,77,78,79]. Different extraction conditions were reported like temperature (30–60 °C) [76], pressure (10–37.9 MPa) [79,80], extraction time, (60–240 min) [77,79], CO_2_ flow rate (6.7–56.7 g/min) [77,80], and the use of different types of co-solvents in several compositions with respect to CO_2_ (0.5–12%, *w*/*w*) [79,81] (Table 5).

In SFE, controlling the above extraction parameters is very important to maximize the extraction yield and minimize the operation cost. Understanding the interaction of temperature and pressure is a fundamental part of the SFE, as both affect the physical properties, like viscosity, diffusivity, and density of the solvent, ScCO_2_. For instance, increasing extraction pressure, increases the density of the solvent and the solvating power of the ScCO_2_, which then easily penetrates the sample matrix to facilitate the extraction rate. However, too high a pressure is not good, because it might result in compacting the extraction bed, and it could restrict the flow of CO_2_, reduce the diffusivity of the solvent, and create channels within the extraction bed, which subsequently reduce the extraction yield [82]. The effect of temperature is inconsistent, especially when we are dealing with the solute that is in the form of a single compound. In such cases, increasing temperature increases the volume of the solvent, and decreases the density and solvating power of the solvent, which subsequently reduces the extraction yield. On the contrary, low extraction temperature reduces the vapor pressure of the solute and volume of the solvent, while increasing the density and solvating power, which results in a higher extraction yield. This unique phenomenon is called the “crossover effect”. Depending on the nature of the solute, the crossover region of the compound in the extraction curve varies [83]. This phenomenon might not be applicable for plant material, where the solute is in the form of a crude extract with several kinds of phenolic compounds. Therefore, conducting several extraction experiments is required to understand the effect of the temperature and pressure, and their interaction effect on the overall extraction yield.

Increasing the flow rate of CO_2_, enhances the mass transfer of the solvent into the sample matrix to facilitate the extraction of the phenolic compound. The increase in mass transfer can consequently shorten the time needed for the extraction. However, a very high flow rate could negatively affect extraction efficiency by flowing around the sample matrix and limiting the mass transfer [84]. Most of the time, SFE of phenolic compounds is conducted in the presence of co-solvents like ethanol, because the solubility of phenolic compounds in ScCO_2_ is low, compared to nonpolar compounds. Enma et al. [79] observed an increase in the yield of phenolic compound extraction from *Sargassum muticum* by 1.5 times, as compared to pure ScCO_2_, when they used ethanol as a co-solvent at 10% (*w*/*w*). They also investigated other solvent flow rates (ranging from 0.5–10%), but only 10% of ethanol showed the highest result. Therefore, careful selection and optimization of the solvent flow is important.

SFE is used less frequently for the extraction of marine phenolics from seaweeds, compared to other emerging technologies reviewed in this paper, and the number of publications is very limited. However, the technology is used extensively for the extraction of phenolic compounds from plant materials. More details and a comprehensive review on the application of supercritical fluid for extraction of phenolic compounds from several types of terrestrial plant materials, were presented by Katarzyna at al. [89]. Some information in these studies could be extrapolated for optimizing the extraction of marine phenolics.

### 3.6. Other Emerging Technologies

The need for an efficient, green, and sustainable method to recover plant and marine bioactives has encouraged researchers to develop several new techniques, with some of them still in development stages. These techniques are based on the energy sources they use for extraction, such as pulsed electric field extraction (PEF), ohmic heating [90], and the use of a centrifugal field for the case of centrifugal partition extraction (CPE) [81]. Based on the use of surfactants to facilitate the extraction (SME) and based on the type of extraction medium they use, such as extraction using a new type of “designer solvents”; including, ionic liquids (IL), deep eutectic solvents (DES), and natural deep eutectic solvents (NDES).

In PEF applications, high voltages (kV range) are applied in pulses of short duration (nano or micro-seconds) with the main objective of causing electro-permeabilization and destroying the cell membranes to accelerate the extraction rate [91]. PEF is used extensively for the extraction of phenolic compounds from terrestrial plants [92,93,94]. There is very limited information on the application of PEF marine resources, however, since the method is proven to be effective for the extraction of phenolic compounds from land plants, it should also be replicated for the extraction of marine phenolics, including seaweeds.

CPE is a multi-stage liquid–liquid extraction technique conducted under a centrifugal field. The extraction of the specific components is based on its partition coefficients between the two liquid phases [95]. This technique is used most frequently for the purification of phenolic compounds from marine resources [96,97,98]. However, Anaëlle et al. [81] used CPE to extract bioactive phenolic compounds from brown seaweed *Sargassum muticum*, and compared the yield and activity of the extracts with two green techniques, SFE and PLE. The result of their study showed that the total phenolic content of the CPE extract was higher than both the PLE and SFE extracts. The concentration of the total phenolic compounds in CPE was twice that of PLE, which gave the second-highest concentration. This study indicated that more studies are needed to explore such kind of alternative techniques, optimize the operation conditions, and apply to other seaweed species.

The use of surfactants in surfactant-mediated extraction (SME) is also a promising, newly emerging technology for the extraction of phenolic compounds [99]. Surfactants can form monomolecular layers on the surface of a liquid, decreasing the interfacial tension between two liquids, allowing the miscibility of two liquid. This could enable SME to be used in the isolation of compounds with a wide range of polarities and complex chemical structures [100]. Yılmaz et al. [100] have conducted a comparative study of SME with EAE and PLE, for isolation of total phenolic compounds and phlorotannins from brown seaweed *Lobophora variegata*. They reported that the yield of both total phenolics and phlorotannins were higher for SME, than EAE and PLE.

Ionic liquids (ILs) are types of simple molten salts, containing a relatively large organic cation and an inorganic anion, which are liquid at or near room temperature. Compared to common organic solutions, ionic liquids have potential advantages like a low melting point, broad liquid temperature, negligible vapor pressure, and extended, specific, solvent properties [101]. In recent years, ILs-based extraction techniques were used for the extraction of phenolic compounds from plant materials [102,103,104]. However, the use of ILs-based extraction of phenolic compounds from the marine resources like seaweeds is scarce [86,105].

Deep eutectic solvents (DESs) as a system formed from a mixture of two or more Lewis acids and bases or Brønsted-Lowry acids and bases that has the lowest freezing point, compared to its starting constituents. The formation of DES results from the complexation of a halide salt, which acts as hydrogen-bond acceptor and a hydrogen-bond donor (HBD). The physical structure of some DESs is thought to be similar to that of the ILs. However, generally, DESs are different in terms of the source of the starting ingredients and the chemical formation process. Hence, the applications of their chemical characteristics are different in many ways [106]. DESs were used extensively for the extraction of phenolic compounds from terrestrial plants [107,108,109,110]. They have already found an application in the extraction of hydrocolloids from the red seaweed *Kappaphycus alvarezii* [111]. However, unlike ILs, to the best of our knowledge, there is no single published article on the use of DESs for the extraction of marine phenolics. Therefore, as these solvents are considered “green”, more research is needed to utilize the potentials of their unique properties for the extraction of phenolic compounds from marine resources for a sustainable future.

### 3.7. Combination of Different Emerging Technologies

A combination of two or more emerging technologies was applied to enhance the extractability of marine phenolics from seaweeds. Thus far, several researchers reported the use of a combination of the emerging technologies, MAE with PLE, UAE with EAE, MAE with EAE, EAE with SFE, and a combination of UAE with MAE. In a recent study, Garcia-Vaquero et al. [40], combined UAE with MAE, for the extraction of bioactive compounds from *Ascophyllum nodosum.* They reported a 30.3% and 10.2% increase of phenolic compounds yield, when a combination of the two methods was used at the same time, as compared to UAE and EAE alone, respectively.

A combination of two different extraction methods might not always result in the high content of phenolic compounds. Sánchez-Camargo et al. [48] pretreated *Sargassum muticum* with different carbohydrases and proteases, to evaluate the effect of phlorotannins extraction, using PLE. The total phenolic contents of all PLE extracts, which were pretreated with enzymes, were lower than the extracts obtained with PLE alone. As described by the authors, the possible reason for the lower result in the total phenolic content of enzyme-pretreated samples is that when the algal cell wall is degraded by proteases or carbohydrases, intracellular contents like proteins are released in the extraction medium. The released proteins bind to phenolic compounds to form complexes, i.e., polyphenols–protein, which further leads to aggregation and eventually precipitation. Therefore, careful selection of the extraction methods to use them in a combined extraction systems is crucial to minimize the cost of extraction and increase the extraction yield and quality of the extracts.

## 4. Opportunities and Challenges with Emerging Technologies

Although the newly emerging technologies show promising opportunities in terms of increasing the extraction efficiency of phenolic compounds, minimizing the extraction time, improving the quality of the extracts, and minimization of the generation of hazardous waste and its associated impact on the environment, the new technologies also have some challenges, especially in terms of industry applications. If we consider EAE, the most commonly available enzymes are nonspecific enzymes like carbohydrases and proteases, which could result in the co-extraction of other components. This could be solved by using specific enzymes like alginate–lyases, glucuronan–lyases, carrageenases, laminarinases, and agarases, specifically targeting the cell wall polysaccharides; alginate, glucuronic acid, laminarian, and agars, respectively. However, most of these enzymes are under laboratory research stage, produced in lower quantity, and have high prices. Thus, further biotechnological and bioengineering techniques, like cloning, are required to produce these enzymes in large quantity, and at a reasonable price, to use them at an industrial level. Technologies like PLE and SFE require state-of-the art equipment like high-pressure pumps, stainless steel extraction vessels, pipes, fittings, valves, pressure regulators, condensers, and process control systems. Therefore, upscaling these technologies would require very high equipment fixed capital and make the technologies expensive to implement at an industrial level. However, once the equipment is installed, the use of cheap and widely available solvents, like CO_2_ and water, and the possibility of recycling the solvents would make the operational costs relatively low. Moreover, the sustainable nature of these processes creates more opportunities to use them at the industrial level.

Similarly, UAE and MAE also have challenges and opportunists, when we consider them for upscaling, and there are some success as well as failure stories on trials for upscaling for extraction of phenolic compounds from other kinds of biomasses. These success stories can be reproduced here for upscaling of phenolics from marine biomass. The authors refer to the work by Belwal et al. [112], which recently published a comprehensive review of scholarly articles on the scaling up of emerging technologies for extraction process, such as EAE, MAE, PEE, PLE SFE, and UAE. In this review work, several lessons learned from the successes and failures stories were critically evaluated and presented to show the opportunities and possible challenges associated with the non-conventional extraction methods. Nevertheless, the opportunities associated with these technologies far outweighed the challenges, especially considering when their impact on the environment, and this makes them preferable and a more applicable technology for a sustainable future.

## 5. Conclusions

Phenolics are a diverse group of compounds found in seaweeds with several biological activities that attract interest from the food, pharmaceutical, and cosmetic industries. To benefit from the marine organisms and more specifically the seaweeds as a potential source of unique and diverse phenolic compounds, extraction plays a crucial role, especially when it is green. Traditional extraction involves the use of organic solvents and requires a longer extraction time. The newly emerging technologies avoid the challenges associated with conventional extraction methods and are considered to be green. They are not only green, but perform better in terms of maximizing the extraction yield, as compared to the conventional methods. A combination of these methods further increase the yield. These techniques act differently to extract the phenolic compounds, depending on their energy source and the extraction mechanism. Some generate heat during the extraction process, which might degrade the phenolic compounds, some others require further purification and cleaning steps. Some of the emerging technologies were tested and upscaled to extract interesting compounds from terrestrial plants, however, more attention is needed to study the phenolics of seaweeds. In most of the extracts reviewed in this study, the phenolics are reported as the total phenolic content. Information regarding the individual phenolics content of seaweed extracts obtained using emerging extraction technologies is very scarce. Such information is important to understand the influence of each emerging extraction technologies on the individual phenolic contents. Thus, in future, researchers working this area should consider including this important information.

## Figures and Tables

**Figure 1 marinedrugs-18-00389-f001:**
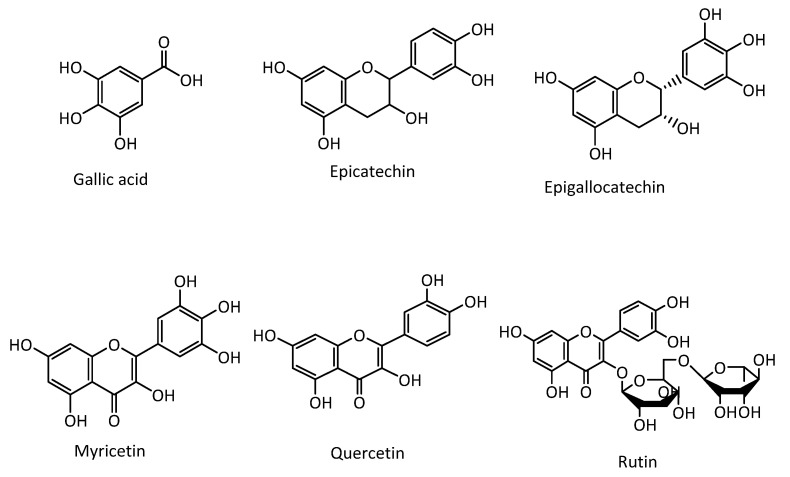
Chemical structures of some commonly reported seaweed phenolic and flavonoid compounds.

**Figure 2 marinedrugs-18-00389-f002:**
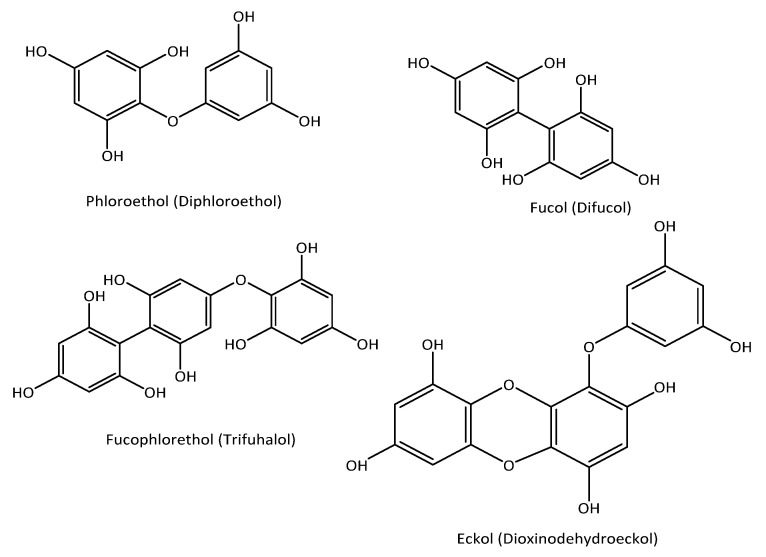
Chemical structures of different kinds of phlorotannins.

**Table 1 marinedrugs-18-00389-t001:** Enzyme-assisted extraction (EAE) of marine phenolics.

Seaweed Type	State of the Seaweed (Wet/Dry/Particle Size)	Type of Enzyme Used	Extraction Conditions Enzyme Conc./Temperature (°C)/Time (min)/pH	Yield (mg GAE/g DW)	Application of the Extract	Reference
*Sargassum boveanum*, *Sargassum angustifolium* *Padina gymnospora*, *Canistrocarpus cervicornis* *Colpomenia sinuosa*, *Iyengaria stellata* *Feldmannia irregularis*	Freeze dried/powdered	Viscozyme AMG 300 L Cellucclast Termamyl Ultraflo L Flavourzyme Alcalase Neutrase	0.1%/50/1200/4.5 0.1%/60/1200/4.5 0.1%/50/1200/4.5 0.1%/60/1200/6 0.1%/40/1200/6 0.1%/50/1200/7 0.1%/50/1200/8 0.1%/50/1200/8	32.4–74.8 * 50.0–84.0 22.3–63.8 26.1–43.0 16.7–32.0 9.5–38.4 28.2–82.5	Antioxidant Antimicrobial	[46]
*Lessonia nigrescens* *Macrocystis pyrifera* *Durvillaea antarctica*	Air dried/powdered (100 µm)	Cellulase α-Amylase	10%/50/1020/4.5	17.38–19.31 21.3 ~13	Angiotensin I-converting enzyme (ACE) activity	[28]
*Enteromorpha prolifera*	Dried/pulverized	AMG 300 L Celluclast Dextrozyme, Maltogenase, Promozyme, Viscozyme Termamyl Alcalase Flavourzyme Neutrase Protamex	2% (*w*/*w*, DW)/-/480/-	2.47 1.5 2.02 1.24 1.83 2.53 2.02 8.44 6.43 6.98 1.83	Antioxidant Anti-acetylcholinesterase Anti-Inflammatory	[47]
*Sargassum muticum*, *Osmundea pinnatifida* *Codium tomentosum*	Oven dried (60 °C)/Powdered (<1.0 mm)	Alcalase Flavourzyme Cellulase Viscozyme L	5% (*w*/*w*, DW)/50/8.0 5% (*w*/*w*, DW)/50/7.0 5% (*w*/*w*, DW)/50/4.5 5% (*w*/*w*, DW)/50/4.5	0.2–0.3 mg CE/g LE 0.1–0.12 mg CE/g LE 0.11–0.16 mg CE/g LE	Antioxidant Antidiabetic	[39]
*Ulva armoricana*	Wet/grounded	Neutral endo-protease A mix of neutral and alkaline endo-proteases A multiple-mix of carbohydrases Mix of endo-1,4-β-xylanase/endo-1,3(4)-β-glucanase Cellulase Exo-β-1,3(4)-glucanase	6% (*w*/*w*, DW)/50/240/6.2	9 11 7 6 4 7	Antioxidant and antiviral	[45]

* When multiple samples are treated with multiple enzymes, the yield of phenolic contents is described in ranges of minimum to maximum values. Further detailed values can be found in the respective references; CE—catechetol equivalent, and LE—lyophilized extracts.

**Table 2 marinedrugs-18-00389-t002:** Ultrasound-assisted extraction (UAE) of marine phenolics.

Seaweed Type	State of the Seaweed (Wet/Dry/Particle Size)	Extraction Conditions Power (W)/Temperature (°C)/Time (min)	Solvent Used	Yield (TPC mg GAE/g DW)	Application of the Extract	Reference
*Ascophyllum nodosum*, *Laminaria hyperborea*	Freeze dried/powdered	750/-/15	Distilled water	0.365 mg PGE/g DW 0.156 mg PGE/g DW	Antioxidant	[65]
*Ecklonia cava*	Far infrared radiation dried (40 °C)/grounded (300 µm)	200/30/720	Water Methanol: water 50:50 Methanol	47.7 63.5 57.9	Antioxidant	[66]
*Laurencia obtuse*	Oven dried (50 °C)/Powdered (1.55 mm)	250/30–50/30–60	95% ethanol	26.23	Antioxidant	[56]
*Hormosira banksii*	Freeze dried/powdered (≤0.6 mm)	150–200/30–50/20–60	70% (*v*/*v*) Ethanol	23.12	Antioxidant	[57]
*Sargassum muticum*, *Osmundea pinnatifida*, *Codium tomentosum*	Oven dried (60 °C)/powdered (<1.0 mm)	400/50/60	Deionized water	235.0 ± 5.57 µg CE/g LE 103.7 ± 1.67 µg CE/g LE 141.1 ± 9.79 µg CE/g LE	antioxidant	[39]

Note: CE—catechetol equivalent, LE—lyophilized extracts; PEG—phloroglucinol equivalent.

**Table 3 marinedrugs-18-00389-t003:** Microwave-assisted extraction (MAE) of marine phenolics.

Seaweed Type	State of the Seaweed (Wet/Dry/Particle Size)	Extraction Conditions Power (W) Temperature (°C)/Time (min)	Solvent Used/Solid:Solvent Ratio (g/mL)	Yield * (TPC mg GAE/g DW)	Application of the Extract	Reference
*Sargassum vestitum*	Freeze dried/powdered (≤600 µm)	720–1200/-/0.42–1.25	Ethanol: water (30–70%)/1:50	58.2	Antioxidant	[62]
*Cystoseira sedoides*	Shade dried/Powdered (200–500 µm)	-/-/0.17–3	Ethanol: water (0–100%)/1:10–1:60	0.38 mg PGE/g DW	Anticancer Activity	[27]
*Ascophyllum nodosum*	Oven dried/Powdered (1 mm)	250,600,1000/-/2–5	0.1 M HCl/1:10	17.9	Antioxidant	[40]
*Chaetomorpha* sp.	Shade dried/powdered (60µm)	200–600/-/4–12	Acetone: water (0–100%)/1:20	0.98 mg TAE/g DW		[64]
*Enteromorpha prolifera*	Shade dried (40 °C)/powdered	300–700/-/5–40 (1–4 cycles)	Ethanol: water (10–60%)/1:10–1:35	0.923	Antioxidant	[73]
*Saccharina japonica*	Dried/powdered (40 µm)	400–600/45–65/5–25	Ethanol: water (50–70%)/1:8–1:12	0.644 mg PGE/g DW	Inhibitory effects on HepG2 cancer cells	[26]
*Caulerpa racemose*	Oven dried (35 ± 2 °C)/Powdered	100–600/20–70/5–60	Ethanol: water (20–100%)/1:10–1:50	6.8	Antioxidant	[61]

* Some of the extraction processes were optimized to get the maximum possible phenolic compounds yield. In such cases, the yield indicated here is the maximum yield. For detailed processes, the readers of this paper are advised to refer to the respective references. PGE—Phloroglucinol equivalent; TAE—Tannic acid equivalent.

**Table 4 marinedrugs-18-00389-t004:** Pressurized liquid extraction (PLE) of marine phenolics.

Seaweed Type	State of the Seaweed (Wet/Dry/Particle Size)	Extraction Solvent	Extraction Temperature (°C)/Pressure (MPa)/Time (min)	Solid: Liquid Ratio (g/mL)	Yield (mg GAE/g DW)	Application of the Extract	Reference
*Sargassum muticum*	Freeze dried/powdered (250 µm)	Ethanol: water (25:75, and 75:25)	120/10.3/20	1:5	101.8	Antioxidant	[81]
*Gracilaria chilensis*	Oven dried (50 °C)/Powdered (0.5 mm)	Water	100/-/5 (3 extraction cycles) 150/-/5 (3 extraction cycles) 200/-/30 (3 extraction cycles)	-	2.06 0.78 10.17	Antioxidant	[85]
*Saccharina japonica*	Freeze dried/Powdered (710 µm)	Water + 0.25 M 1-Butyl-3-methylimidazolium tetrafluoroborate	175/5/5	1:32	58.92 mg PGE/g DW	Antioxidant	[86]
*Laminaria ochroleuca*	Freeze dried/Powdered (<500 µm)	Hexane Ethyl Acetate Ethanol Ethanol: Water (1:1)	80,120,160/10/10	1:20	6 - 83 173.65	Bioactive	[69]
*Fucus serratus*, *Laminaria digitata*, *Gracilaria gracilis*, *Codium fragile*	Freeze dried/powdered	Ethanol: water (80:20) Methanol: water (70:30) Ethanol: water (80:20) Methanol: water (70:30) Ethanol: water (80:20) Methanol: water (70:30) Ethanol: water (80:20) Methanol: water (70:30)	100/6.9/25 90/6.9/25 100/6.9/25 90/6.9/25 100/6.9/25 90/6.9/25 100/6.9/25 90/6.9/25	-	75.96 80.70 1.39 2.93 2.40 0.93 4.76 5.36	Antioxidant	[87]
*Ascophyllum nodosum*, *Pelvetia canaliculata*, *Fucus spiralis* *Ulva intestinalis*	Freeze dried/powdered	Water Ethanol: water (80:20) Acetone: water (80:20) Water Ethanol: water (80:20) Acetone: water (80:20) Water Ethanol: water (80:20) Acetone: water (80:20) Water Ethanol: water (80:20) Acetone: water (80:20)	120/10.3/60 120/10.3/60 60/6.9/60 120/10.3/60 120/10.3/60 60/6.9/60 120/10.3/60 120/10.3/60 60/6.9/60 120/10.3/60 120/10.3/60 60/6.9/60	-	70.4 mg PGE/g DW 66.26 155.95 41.13 40.07 168.82 90.79 124.30 204.40 33.75 20.95 48.56	Antioxidant	[88]

**Table 5 marinedrugs-18-00389-t005:** Supercritical CO_2_ extraction (SFE) of marine phenolics.

Seaweed Type	State of the Seaweed (wet/dry/particle size)	Co-Solvent Used/Co-Solvent Flow Rate (mL/min/%CO_2_ flow rate)	Extraction Conditions Temperature (°C)/Pressure (MP)/CO_2_ Flow Rate (g/min)/Time	Yield (mg GAE/g DW)	Application of the Extract	Reference
*Gracilaria mammillaris*	Vacuum oven dried (45 °C)/Powdered (0.15–0.6 mm)	Ethanol (2–8%, *w*/*w*)	40–60/15–30/6.7/240	3.79	Antioxidant	[77]
*Sargassum muticum*	Freeze dried/powdered (250 µm)	Ethanol (12% *w*/*w*)	60/15.2/-/90	34.5 mg PGE/g DE	Antioxidant	[81]
*Sargassum muticum*	Freeze dried/Powdered (<0.5 mm)	Ethanol (0.5–10%, *w*/*w*)	30–50/10–30/25/60	-	Antioxidant	[79]
*Undaria pinnatifida*	Freeze dried/Powdered (500 µm)	Ethanol/2	30–60/10–30/28.17/60	-	-	[76]
*Laminaria digitata* *Undaria pinnatifida* *Porphyra umbilicalis* *Eucheuma denticulatum* *Gelidium pusillum*	Dried/Powdered	-	50/37.9/56.7/120	23 mg GAE/g DE 4 3 2 15	Antifungal	[80]

PGE—phloroglucinol equivalent; and DE—dry extract.

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
