# Peer review of "Emerging Technologies for the Extraction of Marine Phenolics: Opportunities and Challenges"

_marinedrugs, 2020, doi:10.3390/md18080389_

Round 1

Reviewer 1 Report

The manuscript is an interesting review article concerning different non-conventional methods of the extraction of valuable bioactive phenolic compounds from marine organisms such as several seaweed (macroalgae). Nowadays there is a great interest for new bioactive/pharmacoactive compounds and also a great demand for the development of sustainable technologies, especially regarding the environment pollution.

According to my opinion the manuscript should be accepted for publication, after minor revision.

The comments are as follow:

  • In Figure 1 – the structure denoted as quercetin is quercetin glucoside.
  • It should be interesting if authors make some distinction between phenolic compounds specific for marine organisms, and others which are widespread, in the introduction section.
  • In Table 2 – is it necessary to differentiate among distilled and deionized water?
  • The sentence in lines 322,323 seems to be odd: “solute that is in the form of a single compound”
  • Is there some evidence with a comparison of the extraction yield of the different extraction methods?
  • Reference 80 is incomplete.

Author Response

In Figure 1 – the structure denoted as quercetin is quercetin glucoside.

Response: Thanks for the comment. The structure of quercetin has been corrected in the revised manuscript.

It should be interesting if authors make some distinction between phenolic compounds specific for marine organisms, and others which are widespread, in the introduction section.

Response: Thanks for the comment. Most marine phenolics are similar with other phenolics widely spread in nature. However, some phenolics are specific to marine environment and specific type of seaweeds. For instance, phlorotannins are specific to marine brown seaweeds. This information has been indicated in the introduction section (Line 48-49)

In Table 2 – is it necessary to differentiate among distilled and deionized water?

Response: Thanks for the comment. We have used the type of water as it is mentioned in the original article. Distilled water may sometime contain some dissolved ions in that sense it may differ from deionized water.

The sentence in lines 322,323 seems to be odd: “solute that is in the form of a single compound”

Response: Thanks for the comment. In supercritical fluid extraction, the compound or group of compounds of interest to be extracted by the fluid is solute and the supercritical fluid is solvent. Thus in this case is the solute is a single compound not a group of compound. We hope it is clear now.

Is there some evidence with a comparison of the extraction yield of the different extraction methods?

Response: Thanks for the comment. The comparison of the yield of phenolic compounds has been made based on the dry weight of the seaweed materials used to extract the phenolics throughout the manuscript. Therefore, we may consider this as a common comparison ground for the extraction yield.

Reference 80 is incomplete.

Response: The reference has been corrected in the revised manuscript.

Reviewer 2 Report

The manuscript is interesting but could be improved with more assertive critical analysis. Essential information to evaluate the effectiveness of the extraction, is missing in the tables. This is the state of the seaweed (dried/raw form) and the particle size of the seaweed (Was it milled? Which is the particle size).

Minor comments:

Abstract: “Newly” is not appropriate for the title. Identify the technologies in the abstract.

L65         Correct “befit”.

L136      Realize or accomplish?

L342       A correct flow rate must be given by unit time.

Author Response

The manuscript is interesting but could be improved with more assertive critical analysis. Essential information to evaluate the effectiveness of the extraction, is missing in the tables. This is the state of the seaweed (dried/raw form) and the particle size of the seaweed (Was it milled? Which is the particle size).

Response: Thanks for this very interesting comment. We have included additional column in all tables to indicate the state of the seaweeds used for the extraction of the phenolics. The added column in the tables has been indicated by red color font.

Minor comments:

Abstract: “Newly” is not appropriate for the title. Identify the technologies in the abstract.

Response:  Thanks for the comment. We have deleted the word “Newly” from the title and we have listed some of the technologies in the abstract (Line 19-21).

L65 Correct “befit”.

Response:  Sorry, for the mistake and this has been corrected in the revised manuscript.

L136 Realize or accomplish?

Response:  Thanks for the comment. We have replaced the word “realize” with “accomplish” in the revised manuscript.

L342 A correct flow rate must be given by unit time.

Response: Yes, it is true flow rate must be given by unit time. Here the flow rate is expressed as percentage of the flow rate of CO2 and the flow rate of CO2 was expressed by unit time. This information has been included in the revised manuscript.

Reviewer 3 Report

Regarding the manuscript "Newly Emerging Technologies for Extraction of
 Marine Phenolics: Opportunities and Challenges" I need to constant that this is very interesting area of research especially because marine macroalgae and all research related to blue biotechnology are very wellcome. BUT  the main drawback of this paper, which needs to be completely improved is the part related to INDIVIDUAL PHENOLICS and iTS CONCENTRATION (such as phlorotannins, gallic acid, epicatechin, rutin etc.) obtained in extracts of macroalgae  with each innovative extraction method (EAE,UAE,MAE,PLE,SFE). The main topis of this article is related to phenolics in marine macroalgae and  in all Tables  in paper (Tables 1-5) is just given TOTAL PHENOLIC CONTENT (TPC). This cannot be accepted, without deep discusion and new tables with data about INDIVIDUAL PHENOLS obtained with each extraction method. Also with just pure supercritical CO2 it is not possible to obtained polar compounds such as phenolics, so also explain it.

Please revised completely manuscript and add new tables related to iNDIVIDUAL Phenolics obtained with each extraction method and after that the manuscript can be considered for publication.

Author Response

Regarding the manuscript "Newly Emerging Technologies for Extraction of
 Marine Phenolics: Opportunities and Challenges" I need to constant that this is very interesting area of research especially because marine macroalgae and all research related to blue biotechnology are very wellcome. BUT  the main drawback of this paper, which needs to be completely improved is the part related to INDIVIDUAL PHENOLICS and iTS CONCENTRATION (such as phlorotannins, gallic acid, epicatechin, rutin etc.) obtained in extracts of macroalgae  with each innovative extraction method (EAE,UAE,MAE,PLE,SFE). The main topis of this article is related to phenolics in marine macroalgae and  in all Tables  in paper (Tables 1-5) is just given TOTAL PHENOLIC CONTENT (TPC). This cannot be accepted, without deep discusion and new tables with data about INDIVIDUAL PHENOLS obtained with each extraction method. Also with just pure supercritical CO2 it is not possible to obtained polar compounds such as phenolics, so also explain it.

Please revised completely manuscript and add new tables related to iNDIVIDUAL Phenolics obtained with each extraction method and after that the manuscript can be considered for publication.

Response: We also strongly agree that including the individual phenolics and their concentration is important and that was in our plan when we start this review study. However, the main drawback of the previously published research works is that the phenolics are reported as crude in the form of total phenolic compounds contents (TPC). There are some papers on profiling of individual phenolic compounds from seaweeds but we could not get enough and appropriate reference to compare such data with respect to innovative technologies. That is why we used total phenolic content (TPC). We believe that profiling and quantification of individual phenolics is important and should be included in future studies. We have added few lines (475-480) in the conclusion part to give emphasis on the importance of this information in future studies.

Regarding pure supercritical CO2 for the extraction of phenolics, yes is it true that it’s difficult to extract polar compounds using pure CO2. However, by tuning the extraction parameters such as temperature and pressure we can modify the polarity of supercritical CO2 so that we can extract moderately polar phenolics. Nevertheless, to maximize the extraction efficiency and the yield of the phenolic compound modifying the solvent with a polar co-solvent such as ethanol is important and this is a widely used technique to extract phenolics using supercritical CO2.

Round 2

Reviewer 3 Report

Dear Authors,

You say in Your response that "We also strongly agree that including the individual phenolics and their concentration is important and that was in our plan when we start this review study. However, the main drawback of the previously published research works is that the phenolics are reported as crude in the form of total phenolic compounds contents (TPC). There are some papers on profiling of individual phenolic compounds from seaweeds but we could not get enough and appropriate reference to compare such data with respect to innovative technologies. That is why we used total phenolic content (TPC). We believe that profiling and quantification of individual phenolics is important and should be included in future studies." So according to that the chosen topic of this review paper is not adequate. Your paper is titled "Newly Emerging Technologies for Extraction of Marine Phenolics: Opportunities and Challenges" and You just talk about Total phenolics. So completely new title should be given to this paper.